# Mitochondria-Endoplasmic Reticulum Crosstalk in Parkinson’s Disease: The Role of Brain Renin Angiotensin System Components

**DOI:** 10.3390/biom11111669

**Published:** 2021-11-10

**Authors:** Tuladhar Sunanda, Bipul Ray, Arehally M. Mahalakshmi, Abid Bhat, Luay Rashan, Wiramon Rungratanawanich, Byoung-Joon Song, Musthafa Mohamed Essa, Meena Kishore Sakharkar, Saravana Babu Chidambaram

**Affiliations:** 1Department of Pharmacology, JSS College of Pharmacy, JSS Academy of Higher Education & Research, Mysuru 570015, Karnataka, India; tuladharsunanda4@gmail.com (T.S.); bray365@gmail.com (B.R.); ammahalakshmi@jssuni.edu.in (A.M.M.); abidpharma8088@gmail.com (A.B.); 2Centre for Experimental Pharmacology and Toxicology, JSS Academy of Higher Education & Research, Mysuru 570015, Karnataka, India; 3Biodiversity Research Centre, Dohfar University, Salalah 2059, Oman; lrashan@du.edu.om; 4Section of Molecular Pharmacology and Toxicology, Laboratory of Membrane Biochemistry and Biophysics, National Institute on Alcohol Abuse and Alcoholism, 9000 Rockville Pike, Bethesda, MD 20892, USA; wiramon.rungratanawanich@nih.gov (W.R.); bj.song@nih.gov (B.-J.S.); 5Department of Food Science and Nutrition, CAMS, Sultan Qaboos University, Muscat 123, Oman; Drmdessa@squ.edu.om; 6Ageing and Dementia Research Group, Sultan Qaboos University, Muscat 123, Oman; 7College of Pharmacy and Nutrition, University of Saskatchewan, Saskatoon, SK S7N 5A2, Canada

**Keywords:** ER stress, mitochondrial dysfunction, mitochondrial-associated membrane (MAM), ER–mitochondria crosstalk, brain renin angiotensin system

## Abstract

The past few decades have seen an increased emphasis on the involvement of the mitochondrial-associated membrane (MAM) in various neurodegenerative diseases, particularly in Parkinson’s disease (PD) and Alzheimer’s disease (AD). In PD, alterations in mitochondria, endoplasmic reticulum (ER), and MAM functions affect the secretion and metabolism of proteins, causing an imbalance in calcium homeostasis and oxidative stress. These changes lead to alterations in the translocation of the MAM components, such as IP3R, VDAC, and MFN1 and 2, and consequently disrupt calcium homeostasis and cause misfolded proteins with impaired autophagy, distorted mitochondrial dynamics, and cell death. Various reports indicate the detrimental involvement of the brain renin–angiotensin system (RAS) in oxidative stress, neuroinflammation, and apoptosis in various neurodegenerative diseases. In this review, we attempted to update the reports (using various search engines, such as PubMed, SCOPUS, Elsevier, and Springer Nature) demonstrating the pathogenic interactions between the various proteins present in mitochondria, ER, and MAM with respect to Parkinson’s disease. We also made an attempt to speculate the possible involvement of RAS and its components, i.e., AT1 and AT2 receptors, angiotensinogen, in this crosstalk and PD pathology. The review also collates and provides updated information on the role of MAM in calcium signaling, oxidative stress, neuroinflammation, and apoptosis in PD.

## 1. Parkinson’s Disease

Parkinson’s disease (PD) was first described in 1817 by Dr. James Parkinson, a British physician, as “shaking palsy”. Global epidemiological data reveal that 1–2% of the population belonging to the age group of 65 years and 4–5% at the age of 85 and above are affected by PD, which indicates that ageing is one of the risk factors. Men are 1.5 times more prone to PD than women [1].

The clinical manifestation of PD includes motor dysfunction, namely bradykinesia, hypokinesia, rigidity, resting tremor, postural instability, and non-motor symptoms, such as sleep abnormalities, depression, and dementia. Mechanistically, the dopaminergic degeneration in the substantia nigra pars compacta (SNpc) leads to inhibition of striatal neuronal activities, in turn, causing motor deficits. Similarly, frontostriatal dysfunction leads to attention deficit, disturbance in verbal fluency, and working memory impairment [2]. Dopaminergic degeneration in other brain structures, such as the prefrontal cortex, the hippocampus [3], and the amygdala, are also shown to be involved in PD [4]. Neuronal defects in the locus coeruleus region, wherein a loss of approximately 80% of noradrenergic cell population is observed (which is comparatively more than SNpc dopaminergic loss), are also one of the pathological hallmarks of PD. A reduction in the 5-hydroxytryptamine (5-HT) level has also been linked to a decrease in motor and nonmotor functions and depressive behavior [5].

Accumulation of misfolded proteins is reported as one of the key pathogenic factors that aggravates PD. Several investigations on PD suggest that oxidative stress and bioenergy crisis impair neuronal clearance of misfolded proteins, particularly α-synuclein (α-Syn), leading to neurodegeneration [6,7,8]. Irregular protein conformation of α-Syn, along with intracellular accumulation of Lewy body (LB) and other proteins, such as Aβ and phosphorylated tau (p-Tau), in SNpc dopamine neurons, are frequently reported in the PD post-mortem brains [9]. 

*Parkin* (*PARK 2*), *PINK 1* (*PARK 6*), *DJ-1*/(*PARK 7*), and *ATP13A2* (*PARK 9*) are the major genes in autosomal-recessive PD. On the other hand, *PARK 1*, *α-Syn* (*PARK 4*), *UCHL-1* (*PARK 5*), *LRRK2* (*PARK 8*), *GIGYF2* (*PARK 11*), and *Omi/HTRA2* (*PARK 13*) are the genes involved in autosomal-dominant PD [10]. 

A genome-based study carried out in nine PD patients [11] in Japan and another study [12] in 504 PD patients have shown that *LRRK2* and *α-Syn* are some of the key influencing factors in sporadic PD.

Initially, the renin–angiotensin system (RAS) was identified as an important modulator of the neuroendocrine circuit and involved in the salt and water homeostasis for blood pressure regulation. Eventually, over the last decade, studies reported that the brain RAS also participates in regulating the cerebral blood flow; however, its overactivation is reported to have a pathogenic link in neurological disorders, such as PD, Alzheimer’s disease (AD), transient cerebral ischemia, and depression, and in memory consolidation. In particular, dopaminergic neurons in the brain possess local RAS components, viz., angiotensin II (Ang II) type 1 and type 2 receptors [13]. In the striatum, the expression of angiotensin receptors (AT1/2R) is found in the plasma membrane of spiny neurons. Intriguingly, angiotensin receptors are also located in the nucleus and cytoplasm, as well as in the mitochondrial structure [14].

In the present review, we have summarized the evidence on the participation of the mitochondria and ER in PD pathology, and also the crosstalk between these organelles at the mitochondrial-associated membrane (MAM). Furthermore, mitochondria and ER play a crucial role in oxidative stress, neuroinflammation, and apoptosis, and so does RAS; herein, we also made an attempt to link the possible interactions between RAS–mitochondria–ER at MAM and its impact in PD.

## 2. Endoplasmic Reticulum and PD

ER is the first compartment responsible for synthesis, post-translational modifications, and delivery of proteins to target sites within the secretory pathways and extracellular spaces [15]. Secretory proteins undergo a series of post-transcriptional modifications, including the formation of intra- and intermolecular disulfide bonding and asparagine-linked glycosylation in the ER [16]. Correctly folded proteins are transported to the Golgi complex, while the misfolded proteins are either retained for complete folding or targeted for degradation by autophagy. Derangement in any of these processes causes ER stress, triggering the unfolded protein response (UPR) system. The UPR system is a well-conserved intracellular signaling pathway governed by the activation of various resident ER sensors [17]. Dysfunctional ER leads to oxidative stress and depletion of calcium stores, and vice versa. Mounting evidence indicates that ER stress is the upstream signal for the inflammatory reactions and, thus, involved in the initiation and progression of neurodegeneration. Findings from clinical and preclinical models of PD have revealed that impaired UPR causes an imbalance between cellular damage and ER function [18]. 

The UPR pathway functions to stop misfolded protein translation processes by activating the ER chaperone proteins and attenuating their accumulation in ER. The ER chaperones are responsible for facilitating ER-associated degradation and apoptosis of terminally misfolded proteins [19] (Figure 1B). Glucose-regulated protein or GRP78/Bip, a 78-kDa ERAD E3 ubiquitin ligase, is a major UPR protein that participates in reducing the accumulation of misfolded proteins. GRP78/Bip targets the degradation of mitofusins (MFN 1 and 2) for proteasome degradation through the ER-associated degradation (ERAD) pathway and protects the cells from ER stress. During normal conditions, GRP78 binds and inhibits the activation of the ER response proteins PERK, IRE1, and ATF6. The mechanism of stress-sensing involves the recognition of unfolded proteins by GRP78, which leads to the dissociation from the sensors and releases the repressive interactive proteins [20]. Chronic ER stress activates apoptosis signaling through overexpression of growth arrest and DNA damage-inducible transcript 3 (DDIT3), also known as GADD153 or CHOP [21].

The first study on UPR activation carried out in post-mortem brain samples of PD patients showed a strong immunoreactivity for phosphorylated PERK and eIF2a in the SNpc dopaminergic neurons. It was also observed that neurons having activated PERK had increased levels of α-Syn inclusions [22]. Studies have demonstrated that the stress-responsive proteins, such as HERP, BiP, and phosphorylated protein disulfide isomerase (PDI), are upregulated in PD and co-localized with Lewy bodies. In the MPTP model of PD, protein expression of ER stress markers, GRP78 and CHOP, was reported to be upregulated [23,24]. Credle et al. (2015) demonstrated that α-Syn inhibits ATF6 activity in a cell culture model of PD. Aggregation of α-Syn causes silencing of UPR transcription responses, e.g., the X-box binding protein 1 (XBP1), which, in turn, triggers chronic ER stress, leading to dopaminergic degeneration. HRD1, an ER stress protein or an ERAD-associated E3 ubiquitin protein ligase, helps in ubiquitination and is shown to be localized in the PD neurons [25].

Recent evidence on genetic PD models has indicated a beneficial effect of UPR on the survival of dopaminergic neurons. Although the stress sensor ATF6 is not crucial for the development and survival of dopaminergic neurons, it has been observed that animals deficient in ATF6 have a greater tendency for ubiquitin-positive inclusions, and also dopaminergic cell loss. This suggests that UPR plays an important adaptive function in maintaining protein homeostasis. ATF6 also controls the ERAD component and facilitates astroglial activation in dopaminergic neurons [26]. Another study was performed in a rat model of PD, in which adeno-associated virus (AAV)-mediated expression vectors for Bip and *α-Syn* were co-injected into the SNpc. It was observed that overexpression of Bip significantly reduced synuclein toxicity and improved motor performances due to reduced ER stress [27]. These data suggest that ER homeostasis is maintained by UPR mediators. Accumulation of misfolded proteins for a prolonged period of time leads to failed UPR adaptive responses, which contribute to cell death. Reports also show that PERK/p-eIF2 signaling is increased in the SNpc dopaminergic neurons, which further confirms that PD pathology is strongly linked with the activation of ER stress [22].

## 3. Mitochondria and PD

Mitochondria are present in abundance in tissues with high demand for energy, such as skeletal muscles and the brain [28], and are a major source of free radicals causing cellular oxidative stress. The process of oxidative phosphorylation in the mitochondrial electron transport chain (ETC), along with the production of ATP, is a major source of ROS. However, ROS acts a double-edged sword. Its overproduction causes cellular damage, while the physiological concentration regulates various signaling pathways, including AMPK and MAP kinase pathway [29] and synaptic plasticity [30].

The inner membrane of the mitochondria, called the cristae, comprises of five major enzymatic complexes, namely: complex I (NADH dehydrogenase–ubiquinone oxidoreductase), complex II (succinate dehydrogenase–ubiquinone oxidoreductase), complex III (ubiquinone–cytochrome c oxidoreductase), complex IV (cytochrome c oxidase), and complex V (ATP synthase). These are transmembrane protein complexes that are involved in the ETC and facilitate the movement of electrons from NADH and FADH2 to molecular oxygen. Complexes I, III, and IV are involved in the generation of an electrochemical gradient. However, complex II (succinate dehydrogenase) transfers electrons from succinate directly to quinol and does not contribute to the proton gradient [31]. Several lines of evidence report on the casual relationship between sporadic PD and dysfunctions in the respiratory chain [28,32,33]. The α-Syn monomer is reported to suppress complex I and III activities, whilst α-Syn oligomers increase the intracellular calcium load, and thereby initiate oxidative stress. Abnormal mitochondrial structures and functions are also linked to the PD pathogenic mechanisms. A link between complex I inactivation and PD was established when several groups reported reduced complex I activity in the SNpc of the human brains [34,35,36]. 

In the early stages of PD, elevation in mutated mitochondrial DNA (mtDNA) levels, along with Lewy body formations, is well documented [37]. Alterations in mitochondrial-specific proteins, such as DJ-1, PINK1and Parkin, are also well documented. Mutation in the *Parkin* gene causes protein misfolding and prevents the release of cytochrome C and the opening of the mitochondrial permeability transition pore (mPTP), along with impaired respiratory chain complex I expression [38]. Reduction in the respiratory complex I subunit, along with impaired respiratory capacity in the striatum, leads to oxidative damage and mitochondrial dysfunction [39]. Studies involving knockdown of *Parkin* genes in Drosophila and vertebrate models have shown drastic mitochondrial deficits, including mitochondrial-associated apoptotic muscle degeneration, high vulnerability to oxidative stress, and also an increased sensitivity to mitochondrial toxins, such as rotenone and MPTP [40].

DJ-1 is an ROS scavenger and, during oxidative stress conditions, it translocates to the mitochondria, where it interacts with complex I to maintain cellular homeostasis. During the disease progression in PD patients, peroxisome proliferator-activated receptor gamma coactivator-1α (PGC-1α), a key regulator of mitochondrial biogenesis, was reported to be decreased, leading to death of dopaminergic neurons. In addition, the increase in oligomerization of α-Syn leads to reduced PGC-1α expression and adds up to the neuronal death [41,42]. An imbalance between the mitochondrial fusion and fission has been reported in many neurodegenerative diseases, including PD. Mitochondrial fusion is facilitated by a group of GTPases, namely mitofusins (MFN1 and MFN2), located at the outer mitochondrial membrane, and by OPAI in the inner membrane. Mitochondrial fission requires dynamin-related protein 1 (Drp1) from the cytosol, which assembles into rings to encircle and constrict the mitochondrial tubule [43]. Drp1 is also essential for targeting mitochondria to the nerve terminus, and thus a disruption in the mitochondrial fission can lead to preferential death of nigrostriatal dopamine neurons [42]. Administration of P110, a Drp1 blocker, produced neuroprotection in an MPTP-induced mouse model of PD by inhibiting the p53 apoptotic pathway via Drp1-dependent mitochondrial translocation. Simultaneously, it also suppressed MPTP-induced mitochondrial translocation of proapoptotic proteins, such as Bax and PUMA [44].

An analysis carried out by Reeve et al. has shown that the total mitochondrial count within the axons and the expression of mitochondrial complex I and IV subunits are increased in PD. This suggests that there is an increase in ATP produced within these cells. This increase in mitochondrial volume is due to impaired mitophagy, which leads to a build-up of damaged mitochondria within the synapses [44]. These reports provide evidence that maintenance of mitochondrial functions for neuronal survival is a key target in PD management and can be explored to help improve mitochondrial functions in PD.

## 4. MAM in PD

The concept of the existence of communication between the ER and the mitochondria was initially proposed in 1959 by Copeland and Dalton. The close apposition between the two organelles is formed by a lipid mitochondrial-associated membrane (MAM). The colocalization between the surface of the ER and the mitochondria accounts for 5–20% of the mitochondrial surface [45], and the distance between ER and the outer mitochondrial membrane to the MAM ranges between 10 and 25 nm [46] (Figure 2A,C). The ER domain involved in this association may comprise of either smooth or rough ER. MAM exhibits a few characteristics of the ER, including the activity of enzymes involved in lipid synthesis and the glucose-6-phosphate. However, the specific activity of NADPH dependent Cytochrome reductase, which is a key marker of ER, is only up to 30% in the MAM, when compared to that of the ER [47]. Further, the key markers of mitochondria, the Golgi, and the lysosomes are also significantly present in the MAM [48], along with secretory proteins, such as apolipoprotein B [49]. Mammalian MAM is also rich in lipid biosynthetic enzymes, such as phosphatidylserine synthase (PS), acetyl-CoA: cholesterol acetyltransferase, and diacylglycerol acetyltransferase [50].

The link between the MAM to the ER and mitochondria provides a more functional crosstalk and is important for the movement of biomolecules and crosstalk signaling between the organelles (Figure 1A). It is mainly responsible for the metabolism of phospholipids and cholesterol, and maintains Ca^2+^ homeostasis within the cells. 

Some of the major proteins involved in the crosstalk in MAM include B cell receptor-associated protein 31 (Bap 31), IP3R, a Ca^2+^ channel located in the ER, mitochondrial DRP1, MFN1/2, Fis1, mitochondrial voltage-dependent anion channel 1 (VDAC1), tyrosine phosphatase-interacting protein 51 (PTPIP51), vesicle-associated protein (VAPB), sigma 1 receptors (S1R), enzymes involved in ER redox regulation (e.g., endoplasmic reticulum oxidoreductin 1-α (Ero1-α)), and certain chaperones, such as calnexin and calreticulin. Further, glucose-regulated protein 75 (GRP75), VDAC1, and IP3Rs form a tripartite complex in MAM [52]. This complex between the ER and the OMM helps to regulate Ca^2+^ transfer between ER and mitochondria [53]. Other important MAM-specific proteins include PACS-2 and MFN2, which are involved in the formation and function of the MAM. Downregulation of PACS-2 causes BAP31-dependent mitochondrial fragmentation and leads to uncoupling from the ER surface [54]. BAP31, an ER-specific protein, is mainly involved in protein sorting, and interacts with Fis1 homolog to maintain the MAM tethers [55]. 

MAM plays a crucial role in the survival and death of neurons. These contacts are also involved in shaping of dendritic Ca^2+^ signals and neuronal activity of hippocampal neurons. Importantly, dopaminergic neurons depend on MAM for movement of various metabolites and signaling molecules between these organelles. Initiation of ER stress activates the UPR via GRP78 and caspases and increases the movement of Ca^2+^ to the mitochondria, leading to mitochondrial dysfunction, followed by loss of dopaminergic neurons [56]. Another major evidence by Chan et al. reported that juvenile dopaminergic neurons in the SNpc depend on the L-type Ca (v) 13 Ca^2+^ channels for rhythmic pace-making, which provides a sustained increase in cytosolic Ca^2+^ concentration in the cells. Thus, any alteration in the MAM that affects the movement of Ca^2+^ across the organelles, thereby causing an overload, is expected to pose detrimental effects on the survival of the dopaminergic neurons [57].

Pathological changes in MAM are caused due to mutated PD-related genes, such as *α-Syn*, *DJ-1*, *Parkin*, and *PINK*. As mentioned before, LB pathology, which is composed of aggregates of *α-Syn*, is a prominent feature of PD. The accumulation of LB and mutation in *A53T* genes has shown to increase mitochondrial fragmentation and decrease the MAM association. Further, pathogenic mutation in *α-Syn* leads to a loss of MAM association, along with increased mitochondrial fragmentation and autophagy [58]. Cholesterol present in MAM is critical for the normal functioning of mitochondria and ER, and its alteration is also shown to disrupt autophagy [59]. α-Syn present in MAM influences Ca^2+^ transfer between mitochondria and ER, mediated via IP3 receptor [60]. Similarly, DJ-1 interacts with GRP75 and affects the Ca^2+^ signaling via the GRP75, VDAC, and IP3R tripartite complex [61]. Loss of association between vesicle-associated membrane protein-associated protein B (VAPB) and protein tyrosine-phosphatase-interacting protein 51 (PTPIP51) significantly affects mitochondrial Ca^2+^ reuptake and homeostasis (Figure 2B) [62]. MAM also primarily contributes to the initiation of mitophagy. However, Parkin/PINK1-mediated mitophagy could be a cause for DA neuron death in PD [63]. Mitofusin 2 (MFN2) found on MAM, plays a crucial role in mitochondrial biogenesis by regulating stability at the contact site, Ca^2+^ homeostasis, lipid metabolism, and the structure of mitochondria and ER [64]. The loss of MFN2 also disrupts mitochondria–ER communication, which inhibits mitochondrial Ca^2+^ uptake [65]. Furthermore, depletion of MFN2 affects glucose oxidation, cellular respiration, mitochondrial membrane potential, proton leak, and mitochondrial coenzyme Q levels [66]. Increased expression of Parkin in the MAM fraction due to glutamate excitotoxicity caused ubiquitination of MFN2 and VDAC [67]. Parkin also regulates the ER–mitochondrial tethering, along with PGC1α [68]. These data suggest that the components of MAM and their alterations play a critical role in the pathogenesis of PD and suggest that the increased accumulation of misfolded proteins or mutations in the genes comprising the MAM can be a potential therapeutic target for PD.

## 5. PD-Related Gene Mutations and MAM

### 5.1. α-Syn

Mutation in the *SNCA* gene (which codes for *α-Syn*) is the first reported cause of familial PD [69]. Aggregates of proteins located pre-synaptically, called α-Syn, are found to be localized in the neuronal and the glial cells of the SNpc, hippocampus, thalamus, and the cortex. Within the neurons, α-Syn is present in the cytosol, the synaptic vesicles, mitochondria, and ER [70]. In a study carried out on human and mice brain cell lines, Guardia-Laguarta et al. have reported that wild-type α-Syn was highly enriched in the MAM fraction, rather than in the mitochondria. Further, α-Syn has a greater tendency to attach with high lipid-to-protein ratio molecules, suggesting that it interacts more efficiently with MAM [71]. α-Syn, being a cytosolic protein, has been shown to directly disrupt mitochondrial structure and functions, specifically MAM, which is important for Ca^2+^ homeostasis and apoptosis [72]. Point mutation in *α-Syn* leads to reduction in the association of the protein in the MAM fraction, causing its redistribution to mitochondria or cytosol, along with increased mitochondrial fragmentation [58,73]. 

Overexpression or mutation in α-Syn decreases MAM contacts and consequently leads to disruption in Ca^2+^ homeostasis between the organelles, followed by increased mitochondrial ATP generation. In addition, α-Syn binds to VAPB and decreases its interaction with PTPIP51. Interestingly, this disruption in VAPB–PTPIP51 interaction is seen in neurons derived from familial PD patients harboring pathogenic triplication of α-Syn [74]. A study was conducted using cellular and animal models to observe the behavior of pathogenic point mutations of the *α-Syn* gene at *A53T* and *A30P*. Mutation in these genes leads to a decrease in localization of α-Syn to MAM and an increase in mitochondrial fragmentation. This decreases phospholipid transfer from the ER to the mitochondria [71]. These evidences imply that mutated α-Syn causes alterations in MAM proteins, affecting its composition as well as its functions. 

### 5.2. DJ 1

DJ-1, a multifunctional protein composed of 189 amino acids, located majorly in the cytosol and partly in the mitochondria and nucleus, contributes in maintaining a range of cellular functions, including oxidative stress [75,76]. Human DJ-1 possesses three cystine residues [77], i.e., Cys106, Cys53, and Cys46, of which Cys106 residue acts as a stress sensor [78]. DJ-1 levels were found to be reduced in the SNpc of sporadic PD patients and could be associated with the reduced IP3R3–DJ-1 interaction [79]. Lack of function and mutation of this gene causes decreased mitochondrial membrane potential, and defects in mitochondrial morphology and motility. DJ-1 protein has also been found to be localized at the MAM, where it alters MAM interactions and transfer of Ca^2+^ between ER and mitochondria. In the MAM, DJ-1 interacts with IP3R3, Grp75, and VDAC1 and maintains the flow of Ca^2+^ ions between ER and mitochondria. Ablation of *DJ-1* reduces the MAM association, and dissociation has been reported to be rescued by overexpression of wild-type *DJ-1* but not mutant *DJ-1*.

### 5.3. PINK1 or Parkin 

Parkin is a product of mutation in the *PARK2* gene [80], which encodes for cytosolic E3 ubiquitin protein ligase and is considered as one of the main causes of autosomal recessive PD. Parkin is associated with the quality control pathway that maintains Ca^2+^ transport, mitochondrial biogenesis, fission, fusion, and mitophagy [81]. Various studies using mitochondrial stressors in cells have shown a PINK1-dependent pathway in which Parkin accumulates in the mitochondria, leading to its dysfunction and targeting them to mitophagy. However, its role in maintaining mitochondrial homeostasis, particularly in dopaminergic neurons, needs to be established. 

One of the underlying mechanisms in dopaminergic neuron death is PINK1/Parkin-mediated mitophagy [82], and the initiation site for this process is MAM [83]. A study carried out on neurons exposed to glutamate-induced excitotoxicity has shown that, upon mitophagy, the expression of Parkin is increased in the MAM fraction [84]. Reports have also shown that, in order to initiate the formation of autophagosomes, pro-autophagic proteins relocate to the MAM. In the same context, Parkin-mediated mitophagy can be observed between an MAM fraction with ER and impaired mitochondria [73]. Together with the pro-autophagic proteins Beclin1, Parkin increases ER–mitochondria juxtaposition and promotes autophagosome formation [85]. An increased concentration of Parkin in the MAM fraction also leads to ubiquitination of several MAM proteins, such as VDAC and Mfn2 [67]. Parkin dysfunction and *PARK 2* mutation increase the concentration of Mfn2 in the MAM and downregulate the Mfn2-restored Ca^2+^ fluxes [86].

Similarly, several studies have provided evidence regarding the role of Parkin in the MAM interface. Overexpression of Parkin in the neuroblastoma cells showed an increase in communication between the ER and mitochondria in the MAM fraction and augmented the IP3R-dependent mitochondrial ATP production and transit of Ca^2+^ ions between the organelles [60]. This was also evident in the nigral neurons, where an overexpression of Parkin demonstrated an increase in the contact percentage between the mitochondria and ER to form MAM [68]. Silencing of *PINK1* gene in the M17 dopaminergic cell reduced the number of ER-to-mitochondria contacts and further increased the distance at MAM, and impaired Ca^2+^ homeostasis. In particular, on comparing the fibroblast obtained from patients with mutated *PARK 2* and *PARK 6* genes to that of normal donors, ER–mitochondrial juxtaposition is also reported to be increased [87]. A recent study has also shown that Parkin acts on Mfn2 via MITOL. MITOL is responsible for regulating MAM formation through mitochondrial Mfn2 [88]. Parkin dependent ubiquitination of MITOL is, therefore, responsible for peroxisomal translocation upon induction of mitophagy and helps in modulating the ER–mitochondrial crosstalk at MAM [89].

### 5.4. LRRK2

Mutation in the *LRRK2* gene causes late-onset autosomal PD. Six major mutations in the *LRRK2* genes have been proven to be pathogenic and cause 5–6% of familial and 1–2% of sporadic PD [90]. It is mainly localized in the cytoplasm, as well as in the mitochondria. This large multi-domain protein is involved in a broad range of activities, including cytoskeleton remodeling, protein binding, GTP hydrolysis, microtubule dynamics, autophagy, and maintenance of mitochondrial dynamics [91,92,93]. 

LRRK2 interacts with mitochondrial fission and fusion proteins in both the cytosolic and mitochondrial membrane. *LRRK2* gene mutation causes increased mitochondrial fragmentation and ROS production with decreased ATP generation, thereby leading to increased cellular stress responses [94]. Smith et al. 2016, reported, in a study carried out on PD-derived fibroblasts, that LRRK2 inhibitors decreased mitochondria fragmentation [95]. An overexpression of *LRRK2* gene mutant, particularly *G2019S* and *R1441C*, leads to increased interaction of the mutated protein with Drp1, thereby hyperphosphorylating fission proteins, resulting in increased ROS production and mitochondrial fragmentation [96]. LRRK2 also alters the activity of MFN1, MFN 2, and OPA1, which are proteins involved in regulation of mitochondrial fusion. 

Recent studies have demonstrated the role of LRRK2 kinase by controlling the interaction between the proteins that comprise the MAM. Mutated LRRK2 binds to the E3 ubiquitin ligase, i.e., MARCH5, MULAN, and Parkin, and prevents *PERK*-mediated phosphorylation, while it also prevents the activation of the E3 ubiquitin ligase activity in the MAM. This causes ubiquitination and degradation of the MAM components. Further, this leads to reduced IP3R/VDAC-facilitated Ca^2+^ export and inhibits mitochondrial ATP production. Apart from that, the levels of mitofusin 1 and 2 and PTPIP51 were also decreased as a result of *LRRK2* (G2019S) mutation, thus weakening the MAM interaction [97]. 

## 6. Regulation of Calcium Signaling in MAM

Initial studies by Rizzuto and colleagues emphasized the importance of Ca^2+^ transfer between ER and mitochondria for maintenance of cell architecture, and also provided evidence that the MAM mediates transfer of Ca^2+^ from the ER to the mitochondria [45]. Although Ca^2+^ storage takes place mainly in the ER, Ca^2+^ reserves are also found in mitochondria, particularly in the neurons. The close contact between these organelles provides a low affinity threshold point required for mitochondrial Ca^2+^ uptake [53]. MAM is enriched with proteins, chaperones, and channels involved in handling Ca^2+^. Elevation in intracellular Ca^2+^ occur either due to influx from extracellular spaces through the plasma membrane or due to its release from Ca^2+^ stores located in the ER. Either way, its increase in the ER causes overload of Ca^2+^ in the mitochondria and ultimately leads to cell death [98]. However, the transfer of small fractions of ER-localized Ca^2+^ micro-domains is necessary to sustain a homeostatic communication between the ER and mitochondria [99].

As mentioned above, the movement of the Ca^2+^ micro-domains takes place through a regulated multi-protein complex, comprised of IP3R1, VDAC1, and GRP75 [100]. MAM-associated chaperones, such as calnexin and calreticulin, interact with IP3R and ATPase SERCA-2b to further regulate Ca^2+^ signaling. Increases in Ca^2+^ level in MAM trigger Ca^2+^ influx to the mitochondria through the tripartite complex, which can be reduced by GRP75 knockdown [101]. Investigations report that MAM regulates autophagy by blocking Ca^2+^ transfer to mitochondria through the PTPIP51-VAPB complex [102].

Interruption in the communication between the ER and the mitochondria along with deregulation of Ca^2+^ homeostasis are linked to PD pathogenesis [103]. This indicates that the transfer of Ca^2+^ has to be balanced at the physiological level in order to maintain the mitochondrial functions in the neurons and to prevent cell death [104]. Abnormal increases in Ca^2+^ concentrations are also deleterious to dopaminergic neurons, since elevation in Ca^2+^ may further lead to an increase in DA synthesis due to activation of tyrosine hydroxylase (TH) [105]. This causes intracellular damages due to post-translational modification of α-Syn [106]. The interference of α-Syn with Ca^2+^ homeostasis has been highlighted in a number of experiments. To summarize, the α-Syn remains localized at the MAM and mediates Ca^2+^ transfer by IP3R receptor [60]. An increase in α-Syn oligomerization, particularly by a heterogeneous mixture of small α-Syn oligomers, can be linked to neuronal cell death due to Ca^2+^ dysregulation and mPTP activation [107]. Effect of Ca^2+^ overload on DA neurons can also be initiated due to the activation of calpains. Calpains are a family of Ca^2+^-dependent cystine proteases that are involved in maintaining synaptic plasticity [108]. Overactivation of calpains have been observed in postmortem PD brains, which indicates the effect of dysregulated Ca^2+^ [109]. Further, bioinformatics analysis has shown two single-nucleotide polymorphisms (SNPs) in the genes that encode for CAST, an endogenous inhibitor of the calpastatin gene that leads to the predisposition of idiopathic PD [110].

Our understanding of the architecture and molecular composition of MAM and the relationship between Ca^2+^ signaling and MAM formation and function, although vast, information still remains elusive. Taken together, it is evident that Ca^2+^ regulation plays a vital role in maintaining homeostasis within mitochondria, as well as the ER, thereby affecting the MAM apposition and its functions.

## 7. Role of MAM in Neuroinflammation 

Inflammatory responses in the dopaminergic neurons affect dopaminergic metabolism, decrease antioxidant response, and deregulate calcium responses [111]. Some of the marked features of neuroinflammation in PD include an increase in proinflammatory cytokines and overactivation of microglial cells [112,113,114]. Neuroinflammation takes place in the presence of small molecular motifs or endogenous biomolecules, viz., pathogen-associated molecular patters (PAMPs), damage-associated molecular patterns (DAMPs), etc. These molecules bind to specific host receptors, including toll-like receptors (TLR), nod-like receptors (NLR), and RIG-I receptors (RLR) [115], and provide a self-defensive mechanism against pathogenic stimulation as inflammatory responses.

The first report on the association of MAM and neuroinflammation was established by Zhou et al. in 2011, demonstrating the role of mitochondria in NLRP3 inflammasome activation [116]. NLRP3, an NLR family pyrin, is a protein expressed in macrophages and components of inflammasomes. It is a key proinflammatory pathway involved in neurodegenerative diseases. At resting condition, NLRP3 is found to be localized in the ER. When an inflammatory reaction takes place, along with adaptor apoptosis-associated speck-like protein containing CARD (ASC), it is redistributed in the MAM. NLRP3 can also be triggered by an increase in ROS content or due to ER stress. One of the studies by Zhou et al., 2011 reported that induction of thioredoxin-interacting protein (TXNIP), present in UPR-dependent programmed cell death, can be activated by mitochondrial ROS and translocated to the MAM. This TXNIP induces ER stress via the PERK–IRE1α pathway and activates the NLRP3 inflammasome. Knockdown of the *TXNIP* gene can prevent activation of NLRP3 inflammasome and also IL1β production, thereby preventing inflammation [117]. PERK is an important component of the UPR in ER stress, and also in maintaining MAM juxtaposition to mediate mitochondrial apoptosis [118]. Activation of PERK–JAK1–STAT3 pathway involves microglia and astrocytes and elicits an inflammatory response, causing neuroinflammation. However, this could be abolished by silencing PERK via a specific siRNA [119].

Evidences of neuroinflammation can be seen as compromised BBB integrity and activation of the innate immune system [120]. Neuromelanin (NM) is an iron-binding complex pigment and has been identified in the SNpc and locus coeruleus of human and animal postmortem brain. This complex interacts with accumulated α-Syn and further increases its aggregation [121]. It has been suggested that, when NM is leaked from degenerating neurons, it activates microglia, followed by the release of proinflammatory cytokines, such as IL-6 and TNF-α, and facilitates PD pathogenesis [122,123]. NLRP3 forms a complex with α-Syn and ASC as a response to activation of DAMPs and activates caspase 1, thereby releasing IL-1 β and IL-18 [124]. Further, the immunoregulatory nature of dopamine has also been reported in neuroinflammation [125,126]. The inflammatory responses can contribute to the intrinsic vulnerability of dopaminergic neurons in the SNpc [127]. Brain Ang II via AT1 receptors plays a major role in inducing oxidative stress and inflammation through the NADPH–oxidase complex. Further activation of microglia is also a key process in the progression of PD. In a study carried out on MPTP-induced mice, it is reported that the brain RAS plays an important role in the activation of microglia and that reduction in circulating brain RAS thus reduces inflammation and provides a neuroprotective effect in PD [128]. 

The above data support the activation of glial cells leading to an increase in proinflammatory responses in the case of PD. However, the data on the role of MAM in neuroinflammation and the activation of inflammasomes provide a new area of further exploration.

## 8. Brain RAS

Much evidence has shown a link between the central RAS to that of various neurological disorders. The dopaminergic neurons in the brain possess independent RAS components, including AT1 and AT2 receptors. Auto-radiographic studies have shown the presence of AT1 receptors in dopaminergic neurons of SNpc and striatum in both human and other mammalian neurons [129]. Interestingly, the density of AT1 receptors in the SNpc and striatum is greater in humans as compared to those of rodents and other mammals [130,131]. Overactivation of brain RAS via AT1 receptors has been shown to cause neuroinflammation, oxidative stress, and dopaminergic cell death. This was inhibited on treatment with selective antagonists to AT1 receptor [132]. RAS inhibitors, particularly AT1 receptor antagonists, attenuate dopaminergic degeneration in the MPTP-induced model of PD [133] (Figure 3).

Unlike peripheral RAS, the movement of which is restricted by the BBB, the central RAS is an independent system and plays a vital role in various brain physiological functions and neurological disorders [134]. All the components required for the synthesis of angiotensin peptides are locally synthesized within the brain. The angiotensin converting enzyme (ACE) is an important component of the brain RAS and is predominantly found in the endothelia of the cerebral vasculature, and also in the choroid plexus and astrocytes in the circumventricular structure [135]. Although renin is found in abundance in the blood circulation, its level in the brain is relatively low. It can either exist independently or in the form of pro-renin [136]. Further, the precursor glycoprotein angiotensinogen binds with renin to form Ang I peptides, while ACE further converts the Ang I to Ang II. Evidence has suggested that, in certain instances, the pro-renin binds with the pro-renin receptors instead of binding to renin, leading to angiotensinogen cleavage and overactivation of angiotensin receptors, causing cognitive impairment [137]. Clinical evidences report an increase in ACE activity in the cerebrospinal fluid in PD. Treatment with ACE inhibitors has shown improved motor functions and an increase in level of dopamine [138]. Further, its involvement in cognitive dysfunction was evident, as inhibition of ACE reduced the onset of dementia in AD [139]. After the production of Ang II, its action is mediated by either of the two receptors, i.e., AT1 or AT2 receptors. Both the AT1R and AT2R are G-protein-coupled receptors and are located on the neurons and glial cells of the cortex, basal ganglia, and the hippocampus [140]. The AT1 and AT2 receptor activation generally has opposite actions. While the stimulation of the former cause vasoconstriction, generation of superoxide radicals, and cell proliferation, the latter possesses antioxidant and anti-inflammatory properties [141]. However, in the brain, Ang II increases the expression of AT2R, thus facilitating the RAS activity. Cellular damage in the neurons is reported to be mostly due to the binding of Ang II to AT1 receptors rather than the AT2 receptors and is enabled via the MAPK and JNK stimulation pathways. An abnormal upregulation of Ang II and hyperactivation of AT1R leads to oxidative stress, inflammation, cell death, and thereby, cognitive impairment. Hyperactivation of AT1R can also cause translocation of the AngII–AT1 complex to the nuclease, followed by increased angiotensinogen, renin, and pro-renin receptor binding, ultimately leading to increased brain RAS [142]. 

Mitochondrial AT1R activation has been reported to be the major site of angiotensin-induced ROS production within the neurons [143]. AT1 and AT2 receptors play a major role in regulating oxidative phosphorylation in brain mitochondria. Activation of AT1 receptors in mitochondria facilitates superoxide production via NOX 4 and increases respiration. Mitochondrial AT2 receptors are abundant and induce a decrease in mitochondrial respiration via nitric oxide, modulating oxidative phosphorylation without significant alteration in mitochondrial membrane potential. At the mitochondrial level, AT2 receptors act as respiratory modulators and reduce oxidative stress, which is particularly important in dopaminergic neurons [144].Further, the stimulation of nuclear Ang II receptors enhances mitochondrial biogenesis through PGC1α and increases sirtuin activity, thus protecting the cell against oxidative stress [145].

Alteration in the RAS pathway is also observed over the years in postmortem brain of PD patients. A 90% decrease in AT1 receptor binding in the SNpc and 70% in the caudate and putamen nuclei was observed in human postmortem PD brain tissues [146]. Another study confirmed reduced AT1 immunoreactivity in PD patients when compared to healthy controls and this decrease was attributed to dopaminergic cell loss and diminution in AT1 receptor binding in the neurons [147].

Brain RAS plays a vital role in neuroinflammation and neurodegenerative pathways. Ang II is suggested to trigger neuroinflammation [148,149]. As seen from the literature, it is evident that circulating brain RAS also interacts with ER-specific proteins, causing neuroinflammation, increased ER stress, and neurodegeneration. Another enzyme NADPH oxidase (NOX), found in several regions of the brain, mediates the production of ROS through AT1R-regulated Ang II. The NOX complex causes dopaminergic cell death by stimulating superoxide generation and neuroinflammation [128]. In addition, AT1R activation may also contribute to neuroinflammation, followed by cell death, as evident in PD, due to stimulation of nuclear factor kappa-light-chain enhancer of activated B cells (NF-kB). Moreover, Ang II affects the neuronal N-methyl-D aspartate (NMDA) currents, which leads to an increased NOX-2-dependent ROS production [150]. These results provide evidence for the potential application of AT1R antagonists in the treatment of PD and the potential pleiotropic action of these antagonists as an antihypertensive.

Various evidence has suggested that the neurotoxic effect caused by various neurotoxins, viz., MPTP, 6-OHDA, and rotenone was increased in the presence of Ang II in the dopaminergic neurons [13,132,151]. Further, on treatment with ACE inhibitors and AT1R blockers, there was a significant reduction in the loss of dopaminergic neurons [152]. Inhibition of NADPH oxidase activation also reduced the neuronal loss, suggesting that activation of NADPH and NADPH-derived ROS also contributes to the dopaminergic neuronal death caused by increased Ang II [153]. A study carried out on rats in a chronic hypoperfusion state demonstrated a decrease in dopaminergic neurons and striatal dopaminergic levels, along with increased expression of AT1 receptor when compared to the control group. Treatment with Candesartan, an AT1R blocker, caused attenuation in DA neuron loss, indicating that inhibition of AT1 receptors could be a target for neuroprotection in case of PD [154]. Another prominent study involved an in vitro model of PD using the human neuroglioma H4 cell line with an upregulation of α-Syn. On treatment with Losartan and PD123319, a potential AT2 receptor antagonist, and in the presence of Ang II, a significant reduction in α-Syn-induced toxicity was observed [155]. Previous studies have shown that AT1 antagonists exhibit anti-inflammatory activity via PPARγ. PPARγ, involved in glucose metabolism, energy metabolism, and macrophage differentiation, can be found in neurons and glial cells. Studies have reported that AT1 receptor blockers activate PPARγ, which further inhibits neuroinflammation and renders neuroprotection [156]. This evidence substantiates that the use of angiotensin receptor blockers could offer significant protection in PD by ameliorating neuroinflammation and oxidative stress. 

## 9. Possible Impact of RAS on Mitochondria, ER, and MAM Interaction and Its Link to PD

Convincing evidence in the last decade has focused on investigating the relationship between brain RAS and PD, based on the conclusion that AT1 receptor blockade has a neuroprotective effect on DA neurons. However, targeting the AT2 receptors in PD needs more investigations. Until now, existing data have supported the presence of AT1/2 receptors on the mitochondrial structure, while evidence has also shown that alteration in the brain RAS could combat neurodegeneration caused by ER stress [157]. However, evidence regarding the expression of AT receptors on the ER surface and its correlation with altered RAS remains to be studied. One hypothesis proposes that astrocytes might increase the production and release of Ang II when they are no longer inhibited because of a decreased dopaminergic activity. Ang II could potentially stimulate dopamine release. This could act as a physiological feedback mechanism to keep stable synaptic dopamine levels. It is also clear that excessive Ang II levels result in increased ROS production by the microglia. Elevated ROS will speed up the death of dopaminergic neurons, which would further reduce dopaminergic activity. 

Alterations in MAM functions due to gene mutations, dysregulation in transport of Ca^2+^ within the organelles, increased oxidative stress, and mitochondrial loss are a few validated pathogenic mechanisms linked to PD. Conceptually, it can be expected that, in the DA neurons, ER surface may have the presence of AT1/2 receptors, which yet needs to be studied. Mitigating ER stress caused by impaired UPR functions and protein aggregations could be the possible target in improving MAM function. A more elegant approach, however, would be to understand the role of brain RAS and its direct effect on the MAM and the distribution of AT1 and AT2 receptors on the surface of MAM.

As previously mentioned, mitochondria, being a major source of ROS, participate in the cellular redox signaling. In the neurons, Ang II, Ang III, AT1, and AT2 receptors are colocalized in the inner mitochondrial membrane [144]. Ang II facilitates the generation of mitochondrial ROS, either due to ROS derived from cytoplasmic nitric oxide or due to its direct effect on the mitochondria. AT2 facilitates cytochrome-C release by opening mPTP, and thus triggers mitochondrial-dependent apoptosis in PD [158]. The overproduction of ROS leads to increased oxidative stress and depletes ATP production, which results in DA cell death in PD. In the mitochondria, the presence of AT2 receptors is primarily higher than the AT1 receptors. However, with progressing age, the number of AT1R increases and the activation of mitochondrial AT1R leads to angiotensin-induced ROS production within the neurons, thereby inducing neuronal death [159]. Activation of mitochondrial AT1 leads to an increase in the consumption of oxygen and elevated generation of superoxide via the mitochondrial Nox4. Valenzuela et al. have demonstrated the expression of Nox4 in the mitochondria isolated from SNpc. Interestingly, an increase in oxidative stress within the cells also led to an increase in the mitochondrial AT2 receptor expression to combat the oxidative stress insult. We can thus state that mitochondrial AT1 and AT2 receptors play a crucial role in maintaining the integrity of the organelles [144].

Brain RAS promotes ER stress-associated inflammation and aggravates the neurodegenerative diseases [160]. Blockage of RAS in the brain has been proven to inhibit the ER stress and associated cellular changes. Candesartan, an AT1R antagonist, has been found to reduce the ATF4–CHOP–Puma signaling pathway through inhibition of oxidative stress and prevents the neuronal loss in PD [161]. This signaling, wherein the CHOP binds to the Puma promoter, is a cause of ER stress-induced neuronal apoptosis [162] and, thereby, targeting this pathway by an AT1R antagonist can prevent the induction of PERK-dependent ATF4–CHOP–PUMA signaling and ER stress. Stimulation of AT2R counteracts the obnoxious effects of AT1R activation, thereby preventingneuronal cell death. AT2R stimulation has been found to inhibit NADPH oxidase and reduces oxidative stress in PD [163]. ER is the hub of the protein machinery; reports indicate that AT1R synthesized and assembled in the ER is transported to the Golgi complex for post- translational modifications [164]. Although these data provide a wider arena for identifying AT1/2 receptors on the ER, possibly in DA neurons, the molecular mechanism behind the trafficking of AT1R from the ER to the cell surface remains largely unknown, especially in PD.

Radio-ligand binding studies have given traces of information regarding the existence of AT1R in the MAM isolated from rat liver. However, the existence of AT1R in the MAM could be bystanders, since its presence could be as a consequence of the intracellular trafficking of AT1R present in the plasma membrane via the ER contributionin the presence of MAM. It should, however, be noted that the portion of the ER that is involved in the formation of MAM is flexible with both the MAM surface and the mitochondria; therefore, the AT1R may or may not be present on the MAM surface. This evidence was further validated by Astin et al., who demonstrated that, on treatment with Losartan, an AT1R antagonist, the binding of Ang II to the MAM was inhibited, but remained unaffected when treated with PD123319, an AT2R antagonist [165]. These data reveal the existence of AT receptors on the MAM surface of hepatic tissues providing a strong interlink between RAS and MAM. Based on this evidence we propose to hypothesize the presence of AT1/2R and the RAS components in the MAM fractions in neurons as well, which can prove to be a potential target for various neurodegenerative diseases, including Parkinson’s disease.

Three general approaches are suitable to modulate the deleterious effects of the RAS. The first strategy includes targeting the AT1 receptors present in the mitochondrialmembrane, blocking the AT1 receptors, downregulating the circulating brain RAS, and thus reducing ROS production and mitochondrial dysfunction. The second and third strategies, as we propose, are to establish connections between the ER and MAM, and how the brain RAS affects the crosstalk that exists between these organelles in PD. 

## 10. Limitations

The importance of brain RAS and its impact on ER–mitochondrial crosstalk in the case of PD is an emerging concept. However, after exploring various search engines (PubMed, Google Scholar, Scopus, Web of Science, etc.), abstracting, and indexing databases, no direct evidence could substantiate the existence of brain RAS components at MAM and their relationship in various neurodegenerative diseases.

## 11. Conclusions

MAM provides a platform for crosstalk between the ER and mitochondria, allowing rapid exchange of biological molecules to maintain cellular homeostasis. Prevalence of ER stress and the UPR dysfunction is an important step towards understanding the dopaminergic neuronal vulnerability in PD. Neuronal accumulation of α-Syn causes ER stress and the activation of UPR. Pathogenic mutation in mitochondrial proteins, such as *SNCA*, *DJ-1*, *Parkin*, *PINK1*, *LRRK2*, and *ATP13A2*, is the monogenic cause of familial PD. More recently, novel PD genes, such as *CHCHD2* and *VPS35*, have also been identified to play an essential role in mitochondrial dysfunction. RAS components, viz., AT1R/AT2R, ACE, and angiotensinogen on the said organelles may also play a crucial role in PD. Increase in brain RAS activity and, in turn, oxidative stress due to mitochondrial dysfunction and ER stress is well documented. The present review attempts to gather data on the role of oxidative stress, impaired Ca^2+^ signaling, PD-related gene mutations, and neuroinflammation at MAM and the possible impact of RAS on these pathophysiological aspects. The review also provides hints for the newer research avenues with respect to RAS and MAM in PD.

## Figures and Tables

**Figure 1 biomolecules-11-01669-f001:**
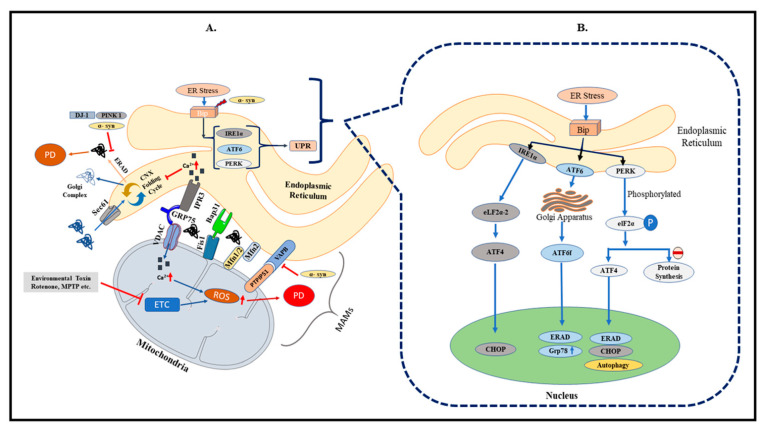
(**A**). An overview of the contact site existing between ER and mitochondria. ER and mitochondria form a close structure that facilitates the movement of Ca^2+^ ion and phospholipids, maintains lipid raft, mitochondrial biogenesis, and regulates cellular homeostasis by apoptosis. Several integral proteins located on MAM include IP3R, VDAC 1, which forms a tripartite with GRP75, Fis1-Bap31 tether, Mfn1 and 2 for maintaining mitochondrial biogenesis, and VAPB–PTPIP51. Pathogenic alterations, either in the mitochondria or ER, lead to changes in the MAM structure and vice versa. Mutation in genes associated with PD leads to aggregation and accumulation of misfolded protein in ER and mitochondria and subsequently increases the distance between MAM. (**B**) The unfolded protein response (UPR) is activated as a response to accumulated proteins within the ER. The UPR consists of membrane resident ER stress sensors, namely PERK, IRE1, and ATF6. Stress conditions due to increased reactive oxygen species (ROS), starvation, and altered Ca^2+^ concentration lead to activation of the UPR. Under normal conditions, GRP78/BiP forms a complex with the three resident proteins. UPR is initiated when a misfolded protein is detected and bound to GRP78, followed by activation of PERK, IRE1, and ATF6. Initiation of UPR and its resident proteins thereby determine the fate of the encountered misfolded protein. During PD, the UPR function is compromised, which causes ER stress, thereby affecting MAM functions.

**Figure 2 biomolecules-11-01669-f002:**
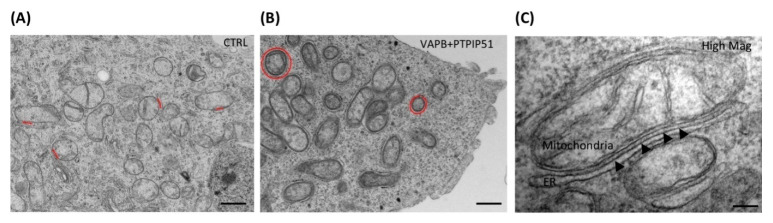
Representative electron microscope image of MAM in NSC34 motor neuron cells [51] (reused as per the Trends in Neuroscience journal’s copyright permission policy under the terms of the Creative Commons CC-BY license). Scale bars = 500 nm (**A**,**B**) and 100 nm (**C**). The figure represents the presence of MAM structures that form a close apposition between ER and mitochondria and the alteration in the cells that might occur as a result of altered protein function within the MAM. (**A**) Depiction of the control cells (CTRL), along with cells transfected with VAPB and PTPIP51. (**B**) ER–mitochondria associations are highlighted in red, demonstrating a drastic increase between MAM associations due to VAPB–PTPIP51 transfection. (**C**) The figure represents a high-magnitude image displaying the close contacts between ER and mitochondria to form the MAM.

**Figure 3 biomolecules-11-01669-f003:**
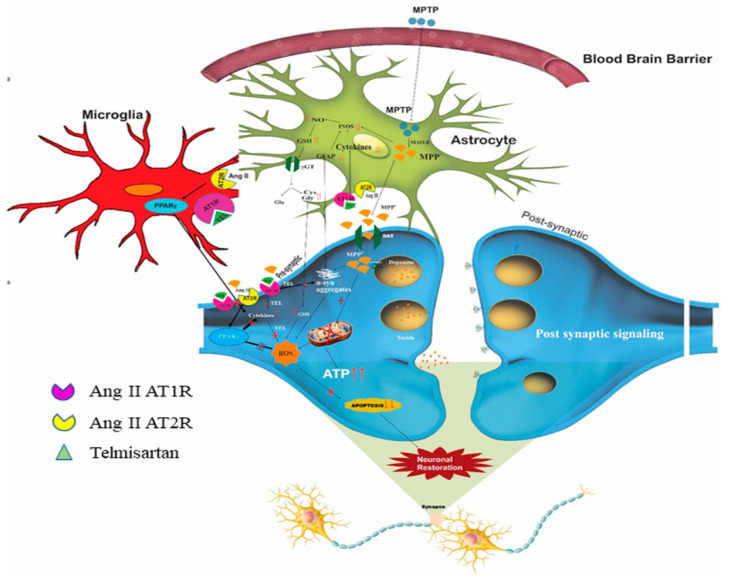
An overview on the distribution of RAS components in neurons, astrocytes and microglia, and the possible mechanism of AT1 receptor antagonist in MPTP induced PD. (Reused as per the copyright guidelines from Springer Nature; Neurotoxicity Research, “Telmisartan Ameliorates Astroglial and Dopaminergic Functions in a Mouse Model of Chronic Parkinsonism”, Sathiya Sekar et al. 2018.(132)). When administered systemically, neurotoxins, such as MPTP cross the BBB and reach the glial cells. MPTP is converted to MPDP^+^ by glial MAO-B, which is subsequently converted to MPP^+^. This toxic MPP^+^ released from the astrocytes enters the DA neurons via the dopamine transporters. Accumulation of MPP^+^ causes neurotoxicity by inhibiting the mitochondrial ETC (complex I), thereby leading to the depletion of ATP, increase in oxidative stress, and, ultimately, neuronal death. The existence of all RAS components, including the angiotensin receptors have been identified in the brain. ATR1/2 and angiotensinogen are expressed in the cell membrane and in astrocytes. An increase in the circulating brain RAS plays a major role in the progression of PD pathophysiology. Therapeutic use of angiotensin receptor blockers provides neuroprotection in various neurodegenerative diseases. TEL, an AT1R blocker, protects the dopaminergic neurons by upregulating TH, DAT, and vesicular VMAT2 expressions, as well as by decreasing oxidative stress and inflammatory cytokine release.

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
