# Peer review of "Mitochondria-Endoplasmic Reticulum Crosstalk in Parkinson’s Disease: The Role of Brain Renin Angiotensin System Components"

_biomolecules, 2021, doi:10.3390/biom11111669_

Round 1

Reviewer 1 Report

Major comments:

In my opinion, the most interesting part of this review is chapter no. 9 “Possible impact of RAS on mitochondria, ER and MAM interaction and its link to PD”. Information given in previous chapters is rather superficial because authors try to cover nearly all possible aspects of Parkinson’s disease in the context of mitochondria, endoplasmic reticulum, mitochondrial-associated membranes and their crosstalk.

I strongly recommend authors to focus on the links between brain RAS components, PD and  Mitochondria – Endoplasmic Reticulum crosstalk.

The title should be specified on the role of the brain RAS system. As a working title I would suggest

Mitochondria – Endoplasmic Reticulum crosstalk in Parkinson’s  disease: the role of brain RAS components

Names of genes should be shown in italics

Statements like “Various evidences have suggested that the neurotoxic effect caused by various neurotoxins viz., MPTP, 6-OHDA and Rotenone was increased in the presence of AngII in the dopaminergic neurons” need citation of corresponding references.

For the paragraph describing the role of RAS in neuroinflammation and neurodegenerative disease (lines 578-590) there are just 2 references.

In the context of this review the authors should provide clear evidence on several important issues: (a) Whether the brain RAS is primarily involved in the development of PD? (b) Where AT receptors are localized? (c) What are the mechanisms responsible for AT receptor signaling relevant for the mitochondria – endoplasmic reticulum crosstalk. 

Minor points: 

English needs editing as the review contains many grammar errors (some examples are given below)

Lines 45-47: “In PD, alterations  in mitochondria, endoplasmic reticulum (ER) and MAM functions affects the secretion”

Line 329: “Mutations in SNCA (which codes for α-Syn) was the first reported”  

Line 578: "Brain RAS play a vital role in neuroinflammation."

Reviewer 2 Report

The manuscript can be accepted for publication since the authors addressed the concern raised by the reviewer

Reviewer 3 Report

Tuladhar et al, present an overview of the important links between mitochondria and the ER in Parkinson's disease. The review follows a logical structure and each section provides comprehensive overviews of key points. The figures included are helpful, but could do with some additional annotation for example in figure 3, mitochondria are also present in astrocytes and microglia. Figure 2 contains images of motor neurons, would dopaminergic neurons not be more appropriate for this review? This review does discuss some important data and theories, however, I do have some concerns. 

1) there are some factual errors within the manuscript. The whole review needs thoroughly checking. For example L-type calcium channels are not mitochondrial, the Reeve paper did not report increased mitochondrial dysfunction within synapses in PD and Parkin is not a product of mutation in PARK2.

2) There are a number of grammatical errors which need correctly and the use of some non-standard phrases which should also be resolved. For example, 'enormous reports' and 'viz' instead of via. 

3) The inclusion of some additional detail of why mitochondria need ATI and 2 receptors is needed.

4) Some sections of the manuscript need attention to the narrative, to ensure they don't read like shopping lists of facts. 

5) All abbreviations should be defined in the text, but should only be defined once and at their first use. 

6) A number of sentences  e.g lines 204-206 say 'a number of groups' or a number of studies, but only have one reference. 

7) line 88-91, the authors list the genes associated with Parkinson's but they are not consistent and some are incorrect. List all genes and include PARK annotations for all. 

Round 2

Reviewer 1 Report

The authors improved the paper and answered most of my queries. 

This manuscript is a resubmission of an earlier submission. The following is a list of the peer review reports and author responses from that submission.

Round 1

Reviewer 1 Report

This manuscript is focused on the relationship between the crosstalk between the mitochondria, ER, and mitochondrial-associated membrane (MAM) and the role of the brain renin-angiotensin system (RAS) in Parkinson's Disease (PD) pathology. Alterations of neuronal cells which lead to the onset and progression of neurodegeneration in PD are linked to dysfunction in the mitochondria, ER and MAM. In particular, disruption or alteration of key cellular process taking place at MAM e.g. calcium homeostasis, mitochondrial dynamics and programmed cell death lead to neuronal degeneration. Furthermore, abnormal up-regulation in the renin–angiotensin system (RAS) function in the brain aggravates neurodegeneration.

The topic of the manuscript is interesting; however, the structure of the paper is not clear. The authors firstly report an overview on PD, following by the role of ER and mitochondria in PD. In the section “The role of MAM in PD” the authors do not really report the evidence of players or functions involved in PD but a simple description of MAM compositions. The section regarding the main proteins, and their mutations, present at MAM linked to the development of PD is interesting and appropriated. Furthermore, the following sections regarding the functions at MAM i.e. oxidative stress, calcium signalling, apoptosis and neuroinflammation are rather out of context since references to PD are sporadic and not completely integrated in the context of the paper (es: point 3). Even the title and the focus of the article should be the renin–angiotensin system (RAS) in PD, and an introductive section regarding the description of the renin–angiotensin system is missing. This may help the readers to better understand the topic. Moreover, it is not easy to understand how the renin–angiotensin system (RAS) is linked to MAM as clear evidence of renin–angiotensin system, MAM and PD interaction are not reported in the present article.

Moreover:

  1. The manuscript contains several typos and grammatical errors, which should be double checked and corrected. For example:

- “In order to maintain the proper folding and translocation of the proteins, newly formed protein passes through the translocon pore (Sec61) and enters the calnexin (CNX) folding mechanism”

- “The co- localization beteween the surface of the ER and the mitochondria accounts for 5-20% of the mitochondrial surface [49], and the distance between the ER and the outter mitochon- drial membrane to the MAM as evident from electron microscopical data (Figure 2A,C) ranges between 10-25mm [50]”.

- “In a sutdy carried out on human and mice brain cell line, Guardia-Laguarta et al., have reported that wild type a-Syn was enriched in the MAM fraction rather than in the mitochondria and point mutation in human a-Syn lead to re- duction in the association of the protein within the MAM fraction causing its redistribu- tion to mitochondria or cytosol along with increased mitochondrial fragmentation [66] [69]”.

- “Another report by Paillusson et al. state that an overexpression or mutation in α-Syn decreases MAM contacts and consequently leads to disruption in Ca2+ homeostasis be- tween the organelles followed by increased mitochndrial ATP generation”.

  1. Also a minor English revision should be performed to better clarify some concepts:

- “When these antioxidants fail to regulate the ROS levels from increasing it causes stress and progressively leads to neurodegeneration [100].”

- “Mitochondria could represent a major source of free radicals resulting in oxidative stress (OS) but at the same time are integral to OS response. It comprises of five major enzymatic complexes embedded in inner mitochondrial membrane namely: complex I (NADH dehydrogen- ase-ubiquinone oxidoreductase), complex II (succinate dehydrogenase-ubiquinone oxi- doreductase), complex III (ubiquinone-cytochrome c oxidoreductase), complex IV (cyto- chrome c oxidase) and complex V (ATP synthase).”

  1. It is not clear the link between VAPB/PTIPIP51/MFN2 and PD in the following paragraph:

“The misfolding of MAM associated proteins or the alteration in MAM functions disrupt cellular structure and physiological functions. This plays a key role in the progression of PD [59]. Loss of association between vesicle-associated membrane protein-associated protein B (VAPB) and protein tyrosine phosphatase interacting protein 51 (PTPIP51) significantly affects mitochondrial Ca2+ reuptake and homeostasis (Figure 2B) [60]. Mitofusin 2 (MFN2) found on MAM, plays a crucial role in mitochondrial biogenesis by regulating stability at the contact site, Ca2+ homeostasis, lipid metabolism and the structure of mitochondria and ER [61]. The loss of MFN2 also disrupts mitochondria-ER communication which inhibits mitochondrial Ca2+ uptake [62]. Furthermore, depletion of MFN2 affects glucose oxidation, cellular respiration, mitochondrial membrane potential and proton leak and mitochondrial coenzyme Q levels [63]”.

What is the evidence linking VAPB/PTIPIP51/MFN2 and PD? This should be reported.

To be considered for publication, the authors should reorganize the structure of the whole paper focusing more on the topic of the review, as the link between RAS, MAMs and PD is not completely clear and only discussed in few sentences at the end of the article.

Reviewer 2 Report

Major comments:

Although the subject of this review sounds interesting “Renin–angiotensin system in mitochondrial-associated membranes and its role in Parkinson’s disease pathophysiology” in the existing form the review does not add useful information for readers, which are not familiar with the problem. In the present form the paper represents a mixture of not well structured literature data in which the role of the brain Renin–Angiotensin System (bRAS) is considered too briefly (just 2 pages). In my opinion, this review should be seriously rearranged and start with more comprehensive characterization of components of the brain Renin–Angiotensin System (bRAS), localization and characterization of specific receptors in mitochondrial and MAM. The references should include a review, which was missed in this submission: Jackson, L.; Eldahshan, W.; Fagan, S.C.; Ergul, A. (2018) Within the Brain: The Renin Angiotensin System. Int. J. Mol. Sci. 19, 876. https://doi.org/10.3390/ijms19030876. Certain attention should be paid to various mechanisms of bRAS signal transduction and the role of mitochondria and ER in this signaling.

Only after this consideration authors should deal with the role of this system in PD.

Other comments:

Authors wrote: “DJ- 1 possess a cystine residue which acts as an oxidative stress sensor”.

First of all correct this sentence as follows: “DJ- 1 possesses a cystine residue which acts as an oxidative stress sensor”. Second, the authors should correct this statement because this protein has three cysteines and only one of them (C106) acts as the oxidative stress sensor and add corresponding references.

Some statements are not supported by references. For example, authors indicate that “wild type DJ-1 comes in association with mitochondrial Hsp70 and increases oxidative stress” [75].

However, in the cited review [75] Hsp70 is mentioned only once and in the different context:

“The interaction of parkin with L166P DJ-1 might involve a larger protein complex that contains chromatin immunoprecipitation assay and Hsp70, perhaps accounting for the lack of parkin -mediated ubiquitination (Moore et al., 2005)”. (see ref. 75, p. 222).

p. 5, Mitochondria and PD: the very beginning of the second paragraph

Authors wrote: “These complexes produce an electrochemical gradient”.

Electrochemical gradient of what? In addition, Complex II is not involved in the mitochondrial electrochemical gradient of protons.

p. 5, the bottom of the page:

The phrase “mitochondrial specific proteins, such as DJ-1” is not correct as DJ-1 may be translocated not only to mitochondria, but also to the nucleus.

There are many typing errors. Some examples are given below:

Page 9, Mutation in DJ1, Line 1

Correct a typing error: 189 amoni acids

The same paragraph, line 4: Correct a typing error: oxidative stress sesor.

Reviewer 3 Report

The authors have explained about the pathophysiology of Parkinson’s disease in detail. This review is definitely summarized about PD from aspects of endoplasmic reticulum and mitochondria, but there are several critical concerns that the authors should address as follows.

Comment 1

There are a lot of wrong letters and misspelled words in this review. I recommend proofreading the review by third-party organization.

Comment 2

It is explained about the relationship during PD, ER and mitochondria in detail in the review. But I think the part of RAS in review is insufficient for explaining the relationship to PD. There are a lot of the explanation of the relationship to AT receptors. The authors should discuss the relationship to angiotensin, renin, and ACE further.

Round 2

Reviewer 1 Report

In my opinion the molecular link between ARS and MAMs is still completely lacking. Only few speculative and vague words on this link are spent in the last section before the conclusions. The title is now even more provocative than before and much less represented in the text. Unfortunately, in my opinion the present review still does not reach the standards to be published. 

Reviewer 2 Report

The authors significantly improved the manuscript. However, it still needs additional work. The authors have added an additional figure 3, but have not described it. In my opinion, they should describe the whole chain of events described in this figure (starting from MPTP accumulation and MPTP conversion catalyzed by monoamine oxidase B etc).  This could be a good summary at least for one model of PD. 

The English text needs careful correction as such sentences given below are confusing: "Parkin is a product of mutation in the PARK2 gene that codes for cytosolic E3 ubiquitin protein ligase" (page 14); "These Lewy body accumulations and mutated A53T genes have shown to decrease the association of MAM and increase mitochondrial fragmentation" (page 9). 

Many abbreviations given in the text have not been explained (e.g. LRRK2). The list of abbreviation could help readers to follow the paper better. 

Reviewer 3 Report

Comment 1

Almost wrong letters and misspelled words have been emended. However, there are still some wrong letters and misspelled words (including space and period) in this review. The authors should revise and confirm.

Comment 2

The authors describe MAM and RAS in great detail. The relationship between these and PD is also explained based on many papers. However, the link between MAM and RAS in the onset of PD is not clear in this review. The authors should explain in detail in this regard.